# Mosquito-Borne Arboviruses Occurrence and Distribution in the Last Three Decades in Central Africa: A Systematic Literature Review

**DOI:** 10.3390/microorganisms12010004

**Published:** 2023-12-19

**Authors:** Natacha Poungou, Silas Lendzele Sevidzem, Aubin Armel Koumba, Christophe Roland Zinga Koumba, Phillipe Mbehang, Richard Onanga, Julien Zahouli Bi Zahouli, Gael Darren Maganga, Luc Salako Djogbénou, Steffen Borrmann, Ayola Akim Adegnika, Stefanie C. Becker, Jacques François Mavoungou, Rodrigue Mintsa Nguéma

**Affiliations:** 1Ecole Doctorale Regionale en Infectiologie Tropical de Franceville (EDR), University of Science and Technique of Masuku (USTM), Franceville P.O. Box 943, Gabon; natachapoungou@gmail.com; 2Laboratoire d’Ecologie des Maladies Transmissibles (LEMAT), Université Libreville Nord (ULN), Libreville P.O. Box 1177, Gabon; 3Département de Biologie et Ecologie Animale, Institut de Recherche en Ecologie Tropicale (IRET-CENAREST), Libreville P.O. Box 13354, Gabon; 4Center of Interdisciplinary Medical Analysis of Franceville (CIRMF), Franceville P.O. Box 769, Gabon; 5Centre d’Entomologie Médicale et Vétérinaire, Université Alassane Ouattara, Bouaké 01 BPV 18, Côte d’Ivoire; 6Université d’Abomey-Calavi, Institut Régional de Santé Publique, Ouidah P.O. Box 384, Benin; 7Institute for Tropical Medicine (ITM), University of Tübingen, 72074 Tübingen, Germany; 8Centre de Recherches Médicales de Lambaréné (CERMEL), Lambaréné P.O. Box 242, Gabon; 9Institute for Parasitology, University of Veterinary Medicine Hannover, 30559 Hannover, Germany; stefanie.becker@tiho-hannover.de

**Keywords:** arboviruses, mosquitoes, epidemics, transmission, Central Africa

## Abstract

Arboviruses represent a real public health problem globally and in the Central African subregion in particular, which represents a high-risk zone for the emergence and re-emergence of arbovirus outbreaks. Furthermore, an updated review on the current arbovirus burden and associated mosquito vectors is lacking for this region. To contribute to filling this knowledge gap, the current study was designed with the following objectives: (i) to systematically review data on the occurrence and distribution of arboviruses and mosquito fauna; and (ii) to identify potential spillover mosquito species in the Central African region in the last 30 years. A web search enabled the documentation of 2454 articles from different online databases. The preferred reporting items for systematic reviews and meta-analyses (PRISMA) and the quality of reporting of meta-analyses (QUORUM) steps for a systematic review enabled the selection of 164 articles that fulfilled our selection criteria. Of the six arboviruses (dengue virus (DENV), chikungunya virus (CHIKV), yellow fever virus (YFV), Zika virus (ZIKV), Rift Valley fever virus (RVFV), and West Nile virus (WNV)) of public health concern studied, the most frequently reported were chikungunya and dengue. The entomological records showed >248 species of mosquitoes regrouped under 15 genera, with *Anopheles* (*n* = 100 species), *Culex* (*n* = 56 species), and *Aedes* (*n* = 52 species) having high species diversity. Three genera were rarely represented, with only one species included, namely, *Orthopodomyia*, *Lutzia*, and *Verrallina*, but individuals of the genera *Toxorhinchites* and *Finlayas* were not identified at the species level. We found that two *Aedes* species (*Ae. aegypti* and *Ae. albopictus*) colonised the same microhabitat and were involved in major epidemics of the six medically important arboviruses, and other less-frequently identified mosquito genera consisted of competent species and were associated with outbreaks of medical and zoonotic arboviruses. The present study reveals a high species richness of competent mosquito vectors that could lead to the spillover of medically important arboviruses in the region. Although epidemiological studies were found, they were not regularly documented, and this also applies to vector competence and transmission studies. Future studies will consider unpublished information in dissertations and technical reports from different countries to allow their information to be more consistent. A regional project, entitled “Ecology of Arboviruses” (EcoVir), is underway in three countries (Gabon, Benin, and Cote d’Ivoire) to generate a more comprehensive epidemiological and entomological data on this topic.

## 1. Introduction

Mosquito-borne arboviruses are viruses that are transmitted by mosquitoes [1]. Arboviruses are maintained in nature by a cycle of biological transmission between a susceptible host and blood-feeding arthropods [2]. The clinical symptoms range from a mild febrile state to severe forms accompanied by haemorrhagic shock or encephalitis [3]. Arboviruses cause a mortality rate of 50% and an estimated 200,000 cases and 30,000 deaths each year, and yellow fever (YF) alone is of major public health concern in Africa and beyond [4]. In addition to YF, dengue (DENV) accounts for over 390 million infections per year [5], and other viruses, such as chikungunya (CHIKV), Zika virus (ZIKV), Rift Valley fever virus (RVFV), and West Nile virus (WNV), have also caused epidemics in Central Africa [6] and are also considered to be arboviruses of major public health concern in this region [7]. These medically important arboviruses have been reported to be silently and actively circulating during major epidemics in Central African countries such as Angola [8], Cameroon [6,9], the Central African Republic [10], Chad [11], Gabon [12], the Democratic republic of Congo [13], Guinea Equatorial [14], and the Republic of the Congo [15]. So far, despite the public health danger of these arboviruses in the subregion, regular surveillance is weak and/or almost absent [16], and some challenges for its non-implementation are as follows: (i) the lack of serological and molecular diagnostic platforms [8]; and (ii) the arbovirus topic is not a priority for research and surveillance for most countries due to other health, security, and political issues [16].

Landscape modifications and climate change as a result of an upsurge in anthropic activities over time in the tropics have modified the natural life cycle and occurrence of major arboviruses and their vectors [17]. Furthermore, in the central African region, the distribution of arboviral disease overlaps with that of competent arthropod vectors, and the increasing risk for its spread has been reported to be associated with deforestation, contact between humans and their wild counterparts, and with local and international transportation across porous terrestrial borders [18]. It appears that tropical species are now invading temperate zones, leading to the emergence of arboviruses that were previously only confined to well-defined regions of the globe. There are more than 3600 species of mosquitoes of the family Culicidae that occurs globally [19]. In the Central African region, two major invasive arboviruses vectors (*Ae. aegypti* and *Ae. albopictus*) have been frequently reported and in some countries of this region, and they are the main drivers of arbovirus spillovers [20], as reported in Cameroon [9].

Although arboviruses and their vectors represents a real public health danger in endemic settings, they are still a neglected topic, and limited studies have been conducted to present updated information on them for the Central African region. To fill this knowledge gap, the present study aimed to conduct a systematic literature review to update the occurrence and distribution of arboviruses of major public health concern, as well as associated spillover vectors in the Central African region, in the last 30 years to guide the control of these viruses and identify avenues for future studies. 

## 2. Materials and Methods

### 2.1. The Study Region

The present study was conducted in eight countries (Angola, Cameroon, Chad, the Central African Republic, the Democratic Republic of Congo, Equatorial Guinea, Gabon, and the Republic of the Congo) of the Central African region. This region is located in the heart of the African continent. Central Africa is home for the second largest tropical rainforest in the world, with annual rainfall that ranges from 100–400 mm per year in the Sahel and over 1600 mm in the tropical rainforests. This ecosystem promotes the survival and proliferation of mosquitoes [1] and the propagation of the diseases they harbour. 

### 2.2. Literature Search Strategy

The preferred reporting items for systematic reviews and meta-analyses (PRISMA, Berlin, Germany) [21] and the quality of reporting of meta-analyses (QUORUM) [22] criteria were applied in paper selection (Figure 1). The search for articles was conducted in the following databases: Google Scholar, Web of Science, PubMed, LILACS, Genesis Library and Z-Library. The following combinations of search keywords were used: “Gabon” OR “Cameroon” OR “Cameroun” OR “Belgian Congo” OR “Democratic Republic of the Congo” OR “Zaïre”, “Republic du Congo” OR “Republic of the Congo” OR “Angola” OR “Tchad” OR “Chad”, “Republic Centre Afrique” OR “Central African Republic” OR “Guinée Equatoriale” OR “Equatorial Guinea”, AND (“Mosquitoes” OR “mosquito-borne virus” OR “arbovirus” OR “arthropod-borne virus” OR “yellow fever” OR “chikungunya” OR “dengue” OR “Rift Valley Fever virus” OR “West Nile virus” OR “Zika” OR “alphavirus” OR “flavivirus” OR “bunyavirus”). Additional information was searched from the World Health Organization (WHO, Geneva, Switzerland) (http://www.who.int, accessed on 21 May 2023) and Centers for Disease Control and Prevention (CDC) (www.ecdc.europa.eu, accessed on 21 May 2023) databases.

### 2.3. Inclusion and Exclusion Criteria

All the papers collected were screened according to the relatedness to the topics of study and the inclusion/exclusion criteria were defined by consensus. The inclusion criteria were as follows: (i) studies conducted in Central Africa; (ii) references from review articles enabled the identification of more relevant papers. The exclusion criteria were as follows: (i) arboviruses detected in animals were not considered but were used in the discussion; (ii) incomplete information such as articles with only an abstract with no relevant information; (iii) studies conducted outside central Africa and reports published before 1993. The reason why we documented the relevant scientific literature from 1993 to 2023 (30 years) was because this period was characterised by major outbreaks in the different countries of Africa [23].

## 3. Results and Discussion 

### 3.1. Papers Selected for the Review

The internet search resulted in the collection of 2454 papers, and only 164 conformed to the stipulated selection criteria (Figure 1). A complete database was created for the 164 relevant articles selected for this study.

### 3.2. Papers Retained by Country

The current study found that the majority of the papers available online on arboviruses and vectors are from Cameroon (58 (35.4%)), followed by Gabon (39 (23.8%)), and the smallest number recorded was from Equatorial Guinea (4 (2.4%)). The details on the distribution of the number of papers published within the study period by country can be found in Figure 2.

### 3.3. The Periodic Trend in Publication of Epidemiological and Entomological Data by Country 

The number of relevant scientific papers on arboviruses and vectors published for the Central African region has improved for some countries and still remains scant for others. We made a graphical presentation of the situation by country, and we found, for Cameroon, that more papers appeared online between 2019 and 2022, with higher entomological studies than epidemiological and both (Figure 3). In Gabon, more publications appeared online between 2020 and 2021 and were dominated by entomological studies (Figure 3). In the Democratic Republic of Congo (DRC), more papers were published between 2018 and 2021 and were rather dorminated by epidemiological studies (Figure 3). In Angola, most papers appeared online in 2006 and were mostly epidemiological studies. For the other countries, such as the Central African Republic (CAR), Chad, Republic of the Congo (RoC), and Equatorial Guinea (EG), both epidemiological and entomological data were sparse and most of them were cross sectional. The low publication turnover for the different countries partly supports the contention that arboviruses are a neglected subject, and the available studies were conducted to confirm cases in suspected outbreak areas in the different countries. The weak trend in publication indicates the lack of regular surveillance of the diseases and vectors and the lack of conjoined and coordinated regional studies.

### 3.4. Occurrence of Arboviruses in the Central African Region

The six arboviruses of public health importance considered in this review belonged to three genera: (i) *Alphavirus* (CHIKV); (ii) *Flavivirus* (DENV, WNV, YFV, ZIKV); (iii) *Phlebovirus* (RVFV). Of these six arboviruses reported to be circulating in the Central African region, CHIKV and DENV have been detected in all the countries of this region [24], and the other four appear to variably occur in the different countries as follows.

#### 3.4.1. Angola

The available and relevant information on arboviruses and associated vectors for Angola was documented in 17 papers. The details on each arbovirus targeted for this review are presented (Table 1) as follows.

##### DENV

Dengue circulation in Angola was first announced in the 1980s by infected travellers returning from Angola to the Netherlands [25]. In 2013, Angola reported its first locally acquired DENV cases [26]. During the 2013 epidemic, about 10% of cases and random cluster participants in Luanda, Angola’s capital city, displayed evidence of recent DENV infection [27]. Furthermore, the genetic study by Neto et al. [8] reported the circulation of DENV2 in Luanda. The detection of dengue during suspect outbreaks or in travelers in Angola was realised using three methods (rapid diagnostic tests, ELISA, and PCR/sequencing). A study conducted in the field used the rapid diagnostic kit (Dengue Duo, Standard Diagnostics) for the detection of the DENV non-structural protein 1 (NS1) (Morbidity and Mortality Weekly Report (MMWR) (http://www.cdc.gov/mmwr, accessed on 21 May 2023) (Ministry of Health Angola and the WHO). The non-structural protein (NSP) ELISA was also used to establish acute infections of DENV [26]. The Trioplex real-time RT-PCR was used to detect dengue, with the advantage that it is capable of differentiating ZIKV and discriminating CHIKV and DENV in coinfected individuals [28]. In addition, the nested PCR was conducted using primers targeting the C-prM region of DENV [29]. There is need for studies on vector competence of local potential mosquito populations for the transmission of DENV in hotspot areas.

##### YFV and CHIKV

In 1970 in Angola, outbreaks of YFV and CHIKV were reported [30]. The first YFV cases in Angola were reported in Luanda. On 13 April 2016, the WHO declared a YFV outbreak in Angola, and during the same period, the WHO also noted a case of RVFV in a man from China working in Luanda, the capital of Angola. Again, in May 2016, a 21 year old female traveller from Luanda to Tokyo tested positive for CHIKV. According to the distribution map proposed by Adam and Jassoy [24], Angola is endemic for YF and CHIKV. Although evidence of circulating YFV and CHIKV was established using high throughput diagnostic techniques in Angola, the issue of the lack of regular surveillance remains. The occurrence of CHIKV was confirmed using the Trioplex real-time RT-PCR [28,31]. The YFV cases were detected using RT-PCR, with primers targeting the 5′ non coding region, and positive samples were further tested using the pan-flavivirus RT-PCR, targeting the flavivirus NS5 gene region using specific primers (FU18993F and cFD29258R [32]. Additionally, although *Aedes*, *Anopheles* spp., and *Culex pipiens* vector compatibility was studied in Angola [33,34], an interesting study was conducted to show the implication of *Aedes* mosquitoes in the transmission of YFV during the 2016 outbreak [35].

#### 3.4.2. Cameroon

All six medically important arboviruses targeted in this review have been reported in Cameroon (Table 1). This study found 58 papers published online on these arboviruses and associated vectors. The situation of the different arboviruses of Cameroon is as follows.

##### CHIKV

In Cameroon, CHIKV was reported in most regions, with high prevalence reported in the northwest (51.4%) [36], and others recorded varing prevalence as follows: Littoral region (12.6–59.4%) [37]; Central region (3–59.4%) [36]; and South West region (4–63%) [38]. However, Cameroon has already been reported as a CHIKV-endemic area with high transmission risk [24]. The 2006, a CHIKV outbreak in Cameroon was confirmed via real-time RT-PCR and partial sequencing of the envelope gene [39]. During the 2006 CHIKV outbreak in the West region of Cameroon, the persistence of anti-CHIKV IgM antibodies was reported in the local population, and entomological studies revealed high relative abundance of *Aedes africanus* [9]. The risk of CHIKV emergence and re-emergence in the west region of Cameroon was supported by an entomological study that showed high species richness and abundance of competent vectors in this part of the country [40]. Interestingly, a study conducted at the border between Cameroon and Gabon showed that patients coming to seek health services in Kyé-ossi in Cameroon from the neighboring town of Gabon (Bitam) were diagnosed positive with CHIKV, indicating the possibility of cross-border transmission between the two countries [18]. The transmission risk of arboviruses at borders between central African countries needs to be studied.

#### DENV

Dengue was found in all the epidemiological studies for Cameroon and was detected in all the 10 regions of the country. The prevalence, by region, or reports of this virus is as follows: Littoral (3.8–68.3%) [41,42,43], Far North (6.7–14.36%) [44,45], West (6.14–14.36%) [45,46], Center (3–45.45%) [45,47], South West (2.5–74%) [6,38] and South (0.5–14.28%). Similar to CHIKV, Dengue is also endemic in the whole of Cameroon with high transmission risk [24]. The nationwide occurrence of DENV has been shown using several diagnostic methods (ELISA and PCR/sequencing). At the regional level, DENV was detected among inpatients using ELISA [48]. In the rural town of Kribi in the south region of Cameroon, MAC-ELISA and the CDC Trioplex real-time RT-PCR were used to show the circulation of DENV [49]. Moreover, the amplification of the partial *E* gene (expected band size of 250 bp) for DENV revealed the occurrence of this virus in the economic capital city (Douala) of Cameroon [41]. The trasmission risk of DENV in this town has already been proven by Kamgang et al. [50] where the two competent vectors (*Ae. aegypti* and *Ae. albopictus*) were identified and their vector roles established for three major towns of Cameroon (Garoua, Douala and Yaounde). Furthermore, a study in the political capital city (Yaounde) of Cameroon revealed high ecological adaptation of *Ae. aegypti* and *Ae. albopictus* and the potential risk for the transmission of arboviruses [51]. The vector ecology and competence of these two vectors of DENV has already been studied in Cameroon [40,50,52,53].

##### YFV

Yellow Fever Virus was detected in two hotspot regions with following prevalences notably: North (25.5%) region [54] and South West region (4–72%) [6,38]. YFV is endemic in Cameroon, with high risk of transmission [24]. It is important to add that the YFV outbreak in Garoua town of the North region of Cameroon was first identified via serology using the MAC-ELISA IgM test, and positive samples were airlifted to the WHO reference laboratory in Dakar, Senegal, for confirmation [54]. The circulation of YFV in this local population is not surprising; Kamgang et al. [50] already identified and established the role of competent *Aedes* vectors in this town. Furthermore, the analysis of laboratory tests results from 2010 to 2020 in Cameroon revealed sustained YFV transmission [55]. It should be noted that Cameroon has a national YFV surveillance system that is planned and implemented by the expanded programme on immunization (EPI) of the Ministry of Public Health, with support from foreign donors.

##### ZIKV

Most of the studies found online concerning the arboviruses of Cameroon frequently reported ZIKV. The prevalence differed in each region as follows: Southwest (11.4%) [6]; Littoral (10–26.2%) [26,44]; East (7.6%) [56]; Far North (2–4.8%) [56]; and Adamawa (2%) [56]. Zika virus has already been reported to be endemic in Cameroon and with high transmission risk [24]. The studies showing the burden of ZIKV in populations of the different regions of Cameroon were conducted using the CDC Trioplex real-time RT-PCR assay, which is capable of discriminating ZIKV from DENV/CHIKV [18]. In fact, it has already been reported via a transmission study that *Ae. aegypti* and *Ae. albopictus* are susceptible to infection and spread of ZIKV in Cameroon [57].

##### WNV

The reports on WNV for Cameroon are scant, and it was only reported in the South West region of Cameoron, with a prevalence range of 3–82% [38]. It is clear from the study of Mayi et al. [40] on vector adaptability conducted in an area bordering the South West region of Cameroon that there was high species richness and diversity of WNV-competent vectors that could represent a transmission risk in the area. However, in the town of Garoua in the North region, a study identified competent vectors of RVFV and WNV [58].

#### 3.4.3. Central African Republic

The CAR has been victim of attacks of 19 arboviruses in the past, and recently, 3 arboviruses were involved in fatal cases: in 1983, WNV was isolated in four patients; two serious cases of YFV occurred in 1985 and 1986; and from 1983 to 1986, RVFV was identified in patients who died from hemorrhagic fever [59]. The details on the different arboviruses are presented (Table 1) in the following paragrahs.

##### CHIKV

Genetic analysis by Tricou et al. [60] confirmed the circulation of CHIKV in the 1970s and 1980s. A serological survey of antibodies to arboviruses was carried out in the human population of the southeast part of CAR in April 1979, and CHIKV was detected to be actively circulating in adult population [61]. The distribution map of CHIKV by Adam and Jassoy [24] shows that CAR is an endemic country for this virus. It is necessary to underline here that CHIKV was frequently studied, and this could be due to the fact that major outbreaks were associated to it. The evidence of circulating CHIKV in CAR was conducted using PCR, where two sets of primers (E1-10145F/E1-11158R and E2-8458F/E2-9240R) were used to amplify the partial sequences of the structural polyprotein gene in the E1 and E2 coding region [62]. Furthermore, an entomological study supported the transmission risk of endemic CHIKV via anthropophilic *Ae. aegypti* and *Ae. albopictus* [63,64].

##### RVFV

An RVFV study conducted in cattle and humans in Bangui reported an overall seroprevalence of anti-RVFV IgM antibodies of 1.9% and that of IgG antibodies of 8.6%. IgM antibodies were found only during the rainy season, but the frequency of IgG antibodies did not differ significantly by season. No evidence of recent RVFV infection was found in 335 people considered at risk; however, 16.7% had evidence of past infection [65]. In another study conducted on cattle, it was found that antibodies to RVFV virus were found in about 8% of adult cattle [66]. The presence of antibodies of CHIKV in cattle indicates their possible role as a reservoir of the disease in Bangui. 

##### YFV, DENV, and ZIKV

In the Central African Republic, since 2006, YFV cases have been notified in the provinces of Ombella-Mpoko, Ouham-Pende, Basse-Kotto, Haute-Kotto, and in Bangui, the capital, which is also an *Aedes* spp.-endemic area. However, the presence of the YFV vectors in the capital city of CAR represents a risk for the spread of the disease. To the best of our knowledge, little or no updated information on YFV, DENV, and ZIKV has been published on the burden of these arboviruses in CAR. However, a distribution map on these three arboviruses of public health concern by Adam and Jassoy [24] showed that CAR is an endemic country for these arboviruses and its vectors. We noticed from the published information on arboviruses of CAR that more entomological studies were conducted to show potential spillover, but epidemiological evidence was scant, probably due to lack of diagnostic capacity and the health care priority being focused on other diseases. An interesting study from 1973 to 1983 in CAR revealed the vector competence status of several species of *Aedes*, *Culex*, and *Anopheles* in the spread of the six medically important arboviruses considered in this current review [67]. Moreover, updated information on competent vectors of arboviruses from 2006 to 2010 was reported [10]. Furthermore, the competent vectors of YFV were identified in CAR [68].

#### 3.4.4. Chad

Three important arboviruses (DENV, YFV, and RVFV) have been reported to occur in Ndjamena (Table 1), the capital city of Chad. Only six papers were found eligible for Chad and were included in the study. The weak publication turnover of Chad could be due to the lack of diagnostic capacity as in most central African countries, where samples are usually sent for further confirmation outside the country, and due to the fact that health care priority is oriented towards other epidemics.

##### DENV

Information on dengue in Chad is not documented and no available evidence on its occurrence was found for the period from 1993 to 2023, but the distribution map of Adam and Jassoy [24] indicates the presence of this arbovirus in this country. The occurrence of competent mosquito vectors have already been reported in the DENV outbreak areas of Chad, and this portrays the risk for the transmission of arboviruses [69].

##### YFV

A low prevalence (0.28%) of YF was obtained from jaundice patients in Ndjamena from 2015 to 2020 during a non-outbreak period [70]. Chad has also been reported to be a yellow-fever-endemic area with high transmission risk [24]. It is important to add that the detection of YFV was carried out by MAC-ELISA-CDC [70].

##### RVFV

This is a zoonotic febrile disease that affects livestock and humans and was first reported in Chad in 1967 [71] and in the same period in Cameroon; since then, this disease has spread beyond the subregion. Apart from this preliminary report, another report by Durand et al. [72] revealed a prevalence rate of 4% in French troops. The evidence of circulating RVFV among these French soldiers was carried out using ELISA and confirmed via real time PCR/sequencing using primers targeting the L, M, and S regions of the genome [72]. In fact, the competent vectors of this arbovirus have already been reported [69].

#### 3.4.5. Democratic Republic of Congo

The available and relevant information on arboviruses and associated vectors for the Democratic Republic of Congo was documented in 19 papers. The details on each arbovirus targeted for this review is presented (Table 1) as follows.

##### YFV

Following the indepence of the DRC in 1960, YF epidemics have been reported in all 26 provinces of the country. A two-year survey (2013 to 2014) reported a YFV prevalence of 31.5% among children in the DRC [73]. From 5 December 2015 to November 2016, a large YF outbreak affected Angola and the DRC, with 7334 suspected cases, of which 962 have been confirmed, and 393 deaths were reported to the WHO as of 28 October 2016 [74,75]. According to the updated distribution map of Adam and Jassoy [24], the DRC is a YF-endemic area with high transmission risk [75,76]. A recent report on YF in the DRC was conducted in Kinshasa, its capital city, with a seroprevalence range of 6–73% [73,77] (Table 1). The occurrence of YFV in the capital city of this country was not astonishing, as it is known that densely populated cities, where high densities of mosquitoes coexists with city inhabitants, are a favouratble milieu for an epidemic of massive proportions. Moreover, the existence of a high density of competent *Aedes* vectors of YFV, already identified in the DRC, is the major driver of major epidemics [78].

##### CHIKV

In the last two decades, Kinshasa, the capital of the DRC, experienced CHIKV epidemics in the years 1999 and 2000, with an estimated 50,000 reported cases [79]. In addition, also in Kinshasa, another outbreak occurred in 2012 [80]. Apart from Kinshasa, other provinces where CHIKV has been reported are Kisangani [81] and Matadi [78]. According to the *Aedes* spp. and the CHIKV distribution maps published in Adam and Jassoy [24], the DRC is a CHIKV-endemic area with high transmission risk [76]. The evidence for the occurrence and spread of CHIKV in the DRC was obtained through high-throughput diagnostic approaches such as PCR/sequencing, where a CHIKV-specific RT-qPCR was performed using primers targeting a 77 bp portion of the non-structural protein 1 (NSP-1), as described by Planning and collaborators [78]. An entomological investigation led to the identification of *Ae. albopictus* as the primary vector of CHIKV [78]. Similarly, another study used primers that rather targeted the E1/3′UTR region, as evidence of the re-emergence of CHIKV in DRC [82]. The principal vector involved in CHIKV transmission in the DRC has been reported to be *Ae. albopictus* [83,84].

##### DENV

Dengue fever virus is one of the common mosquito-borne viruses in the DRC, and cases of this disease has been reported in some hotspot provinces such as Kisangani [81] and Kinshasa [73,80,85,86] via seroepidemiological studies [73,87]. Furthermore, serotyping information on the circulating DENV in the DRC was not available until a survey reported 16 DENV-1 and DENV-2 cases from 2003 to 2012 [88]. Genetic analysis revealed that the DENV-1 strain that caused the 2013 epidemic in Angola also circulated in the DRC in 2015 [89]. Three serotypes of DENV (DENV-1, DENV-2, and DENV-3) have been recorded in the DRC, the most frequent being serotype DENV-1 [73]. In Kisangani, DENV co-circulated with CHIKV during the WNV outbreak of 1998 [81]. In Kinshasa, co-occurrence of dengue and chikungunya was reported during the 2012 outbreak [80]. To conclude, the DENV updated map for sub-Saharan Africa (SSA), published in Adam and Jassoy [24], shows that the DRC is a DENV-endemic area with high transmission risk [76]. The evidence of DENV in the population was confirmed via PCR/sequencing, where the DENV-1 was detected using pan-flavivirus nested RT-PCR with primers targeting the non-structural protein 5 (NS5) gene [89]. Similarly, another study for the confirmation of DENV-1 rather used specific primers targeting the *E* gene [85]. The circulation of DENV-1 was not surprising, as entomological studies reported the presence of competent vectors in the DRC [83,84].

##### ZIKV

Only a few studies present relevant information on the burden and distributon of Zika in the DRC, and only one study reported on its occurrence. A serological study from 2013 to 2014 showed a prevalence rate of 3.5% for ZIKV antibodies in sud-Ubangi [73]. Another study by [88], for the period 2003 to 2011, showed a negative test result for ZIKV using the polymerase chain reaction (PCR) method. The occurrence of Zika virus in the DRC has also been shown in the updated distribution map for ZIKV for SSA [24].

##### RVFV

In 1998, in the Kisangani area of the DRC, RVF has was reported in humans with a low prevalence of 4% [81]. As a zoonotic arbovirus, it has also been reported in domestic animals such as cattle, where in 2009, a seroprevalence of 20% was reported in this animal species in Katanga [90]. Moreover, a seroprevalence rate range of 2–16% among cattle was reported in the Nord-Kivu, Sud-Kivu, and Ituri provinces from the Eastern region [91]. A transmission risk study conducted in 2014 revealed that *Aedes* mosquitoes harbored RVFV [92].

##### WNV

From the available information on WNV of the DRC, only one paper presents relevant information on clinical cases of the disease, and this was in 1998, when a high seroprevalence of 66% was reported in Kisangani [81]. Other information on this virus has been obtained from research on wild animals to establish their potential epizootiological role in its spread to humans, where WNV antibodies were detected in Haut-Uelé Province in chimpanzee [93], bufalo, and elephant in the Garamba National Park [91].

#### 3.4.6. Equatorial Guinea

The available and relevant information on arboviruses and associated vectors for Equatorial Guinea (EG) was documented in only four papers. The weak publication frequency in EG could be due to the lack of diagnostic facilities, as in most central African countries, where samples are mostly sent for further confirmation outside the country, and also due to the fact that the health care priority is oriented towards other epidemics. It is important to add that the use of less sensitive and specific techniques, such as rapid diagnostic tests and ELISAs, could lead to the under-reporting of the occurrence and burden of some key arboviruses in many African countries. The details on each arbovirus targeted for this review are presented (Table 1) as follows.

##### CHIKV, DENV, and YFV

From the best of our knowledge, the only arboviruses that have been reported in EG are CHIKV, DENV, and YFV. Chikungunya was first detected in 2002 and 2003 [94]. In 2006, one of the three travelers returning from EG was diagnosed as positive for CHIKV in Spain [14]. Inaddition, EG is known to be CHIKV-endemic [24]. Similarly, only one study presents the distribution map of DENV and YFV of EG, and it shows that the country is endemic for the two arboviral types [24]. The PCR/sequencing techniques provided evidence for the circulation of CHIKV, where sequences of amplified fragments corresponding to 195 bp of the non-structural protein 4 gene of alphaviruses identified a homogenous cluster of this arbovirus in the 2002 and 2006 outbreaks [14]. In order to obtain a sequence with more phylogenetic information, primers designed by Powers et al. were used to amplify a fragment of a region of the envelope 1 (E1) gene often used for CHIKV phylogenetic analysis [14]. During major arboviral outbreak periods, entomological reports showed the wide spread of *Ae. albopictus* vectors in Bioko in EG [95].

#### 3.4.7. Gabon

In Gabon, arboviruses and associated vectors have been reported in 39 papers. The publication trend has evolved positively for epidemiological studies, with a high number registered in 2022. All six arboviruses of public health concern have been identified (Table 1), but their prevalence and distribution differed with each province as follows.

##### CHIKV

This virus was reported in the whole of Gabon [96], and two important outbreaks occurred in 2007 and 2010 [97] in the Estuaire province of the capital city (Libreville) of Gabon. In Libreville, the prevalence range was 3–86% [98], followed by Haut-Ogooue (45.2–62.3%) [97], and then Ogooue-Lolo (28.7%) [99]. Recent reports on CHIKV are from the Moyen-Ogooue province, with a prevalence range of 0.6–61.2% [6,100]. Chikungunya is endemic in Gabon, with high transmission risk [24]. The evidence of circulating CHIKV in Gabon was confirmed using high-throughput PCR/sequencing using specific primers (OP16 and OP17) [101]. Another study targeted the *E*1 gene for CHIKV, as well as the 3′UTR region, during the 2007 and 2010 CHIKV/DENV outbreaks in Libreville, the capital city of Gabon [99]. It is important to underline that the 2010 arbovirus outbreak in Gabon was driven by CHIKV and DENV [96,100]. This period (2007 and 2010) was characterised by the invasion and wide spread of competent *Aedes* vectors in Libreville [102,103,104].

##### DENV

Similar to CHIKV, DENV in Gabon was identified in all the regions surveyed [96,105] and has been reported to be endemic [24]. The DENV hotspot areas are Moyen-Ogooue (12.3–88.24%) [7,106,107], Haut-Ogooue (12.2%) [97], and Estuaire (4–21.4%) [98]. The evidence of DENV-2 circulation in Gabon by PCR/sequencing was carried out using primers to amplify the envelope (E) gene (758 bp; genome position 1503–2260 nt). Similarly, consensus DENV-1 and DENV-3 PCR fragments of the *E* gene corresponding to a 472 bp fragment of DENV-1 (genome position 1234–1705 nt) and to a 935 bp fragment of DENV-3 (genome position 1256 to 2190 nt) were amplified [105]. Another study was conducted to show the evidence of DENV-3 having amplified the full length of the envelope gene (1479 bp) [106]. Entomological studies found that *Ae. aegypti* and *Ae. albopictus* were associated with CHIKV and DENV-2 [98,108]. In 2021, the re-emergence of DENV, CHIKV, and ZIKV was established via PCR/sequencing using primers targeting the envelope of dengue virus serotype 1 (1485 bp). The updated entomological studies in both urban [109,110] and sylvan environments [111] showed high density and species richness of *Aedes* mosquito vectors in Gabon. Another interesting finding in Gabon was that which identified genes underlying specific resistance of DENV-1 and DENV-3 in *Ae. aegypti* [104].

##### RVFV, YFV, WNV, and ZIKV

The reports on RVFV, YFV, WNV, and ZIKV for Gabon are scant; only one paper reported the presence of these four arboviruses, and only in the Moyen-Ogooue province, with varying prevalences: YFV (60.7%); ZIKV (40.3%); WNV (25.3%); and RVFV (14.3%) [7]. The evidence from serology/RT-PCR tests shows the recent circulation of the six medically important arboviruses considered in this study [7]. The evidence of circulating ZIKV was made through PCR/sequencing, where the non-structural protein 3 (1851 bp) was targeted, and then further screening targeted non-structural proteins (772 bp) and envelope (750) genes of ZIKV [12]. Moreover, the evidence of circulating ZIKV in Gabon was made by amplifying the *E* and *NS3* region using specific primer sequences already published for *E* genes (ZIK-ES1/ZIK-ER1, ZIK-ES2/ZIK-ER2) and *NS3* genes (ZIK-NS3FS/ZIK-NS3FR and ZIK-X1/ZIK-X2), and the entomological part of this study revealed *Ae. albopictus* as the primary vector [20]. It is important to state that the RVFV reported in Gabon was only confirmed via serology and the presence of competent vectors [112].

#### 3.4.8. The Republic of the Congo

The available and relevant information on arboviruses and associated vectors for the RoC was documented in 10 papers. Evidence of the circulating arboviruses in the RoC was based on rapid diagnostic tests, ELISA, and RT-PCR/sequencing. It is known that the sensitivity and specificity of any rapid antibody (IgG or IgM) ELISA or RT-PCR could be of limited value during the initial phase of the transmission window of the disease, as the level of viraemia and IgM antibody titres may be below the limits of detection [113]. The details for each arbovirus targeted for this review is presented (Table 1) as follows.

##### CHIKV

In January 2019, an outbreak of CHIKV fever was reported near Pointe-Noire. This study found a novel CHIKV strain and established the presence of the A226V substitution and close relation with *Aedes aegypti*-associated Central Africa chikungunya strains [114]. Similarly, in 9 February 2019, during the CHIKV outbreak, investigations found two new CHIKV sequences of the East/Central/South African (ECSA) lineage, clustering with the recent enzootic sub-clade 2, showing the A226V mutation. Entomological surveys reported one *Ae. albopictus* pool to be RT-PCR positive [15]. The establishment of the occurrence of CHIKV in the RoC was conducted using two methods (rapid diagnostic test (RDT) and RT-PCR/sequencing). The RDT for specific IgG and IgM detection (STANDARD F Chikungunya IgM/IgG FIA SD BIOSENSOR, Chungcheongbuk, Republic of Korea) was used, and RT-PCR/sequencing was conducted using primers designed by referring to the sequences of the Pakistan-07/2016 CHIKV isolate complete genome [15]. Moreover, the evidence of the CHIKV 2011 outbreak in the RoC was obtained via RT-PCR, where primers previously designed to sequence the LR2006 OPYI CHIKV strain were used to generate PCR products [115]. *Aedes albopictus* was identified to be the primary vector of CHIKV in the RoC [15]. The re-emergence of CHIKV in the RoC was due to the wide spread and dense population of *Ae. albopictus*, as already reported [116].

##### DENV

Although DENV is one of the frequently reported arboviruses in the Central African subregion, to the best of our knowledge, no relevant information has been presented on its burden and occurrence in the RoC. However, the distribution map of DENV for SSA by Adam and Jassoy [24] clearly shows that the RoC is endemic for this virus and its *Aedes* spp. Vectors [116].

##### ZIKV

The lone study reporting the occurrence of ZIKV in the RoC was conducted on 386 serum specimens from volunteer blood donors in 2011 from rural and urban areas of the Republic of the Congo. The result of this study showed a low ZIKV seropositivity rate (1.8%) [117]. The occurrence of competent vectors (*Ae. aegypti* and *Ae. albopictus*) of ZIKV is evidence of risk for its transmission in the RoC [118].

### 3.5. Distribution of Arboviruses in the Central African Subregion

Chikungunya and Dengue were the most frequently detected of the six studied arboviruses of medical importance. Indeed, some countries have already witnessed historic epidemic waves of CHIKV; for instance, in Cameroon in 2006, in Gabon (2007 to 2010), in Congo Brazzaville in 2011, and in the DRC in 2019. The circulating arboviruses in the different Central African countries are presented in Table 1. The widespread distribution of CHIKV and DENV could be attributed to the suitable ecological variables for its vectors, and, of course, it has already been reported that DENV is the most prevalent of all arboviruses [119,120]. Although RVFV cases have already been reported in clinical cases in countries such as Chad, CAR, Gabon, and the DRC, data are still scant, but in a country such as Cameroon, information is mostly available for animal species (cattle, sheep, and goats) [121,122]. The free circulation of livestock and people in the Central African regional corridor could be the main driver of the circulating strains of major arboviruses in countries of this region [6]. The circulation of RVFV in domesticated ruminants in countries of this region could indicate a possible risk of human exposure to zoonotic strains [123]. Moreover, information on WNV was also poorly documented, with clinical cases reported in Cameroon, Gabon, and the Democratic Republic of Congo. Moreover, information on WNV in animals is scant and needs to be documented. The detection discrepancies between countries could be multifactorial, as follows: (i) lack of knowledge; (ii) low or lack of diagnostic capacity; and (iii) poor surveillance systems. The lack of data from the Central African subregion makes it difficult to generate robust and quality field epidemiological and entomological information that could inform us of the patterns and drivers of arthropod-borne diseases transmission [124,125].
microorganisms-12-00004-t001_Table 1Table 1Occurrence and burden of major arboviruses in different countries of the Central African subregion from January 1993 to June 2023.CountrySite (Region, Province, City)ArbovirusDiagnosisProportions (%)ReferencesGabonEstuaireCHIKVP, S, S3–86[97,98,101]DENVP, S, S4–21.4[97,98,101]Moyen OgooueCHIKVS + P, S, S0.6–61.2[7,100,126]DENVS + P, S, S + P, S + P12.3–88.24[7,100,106,107]RVFVS + P14.3[7]YFVS + P60.7[7]WNVS + P25.3[7]ZIKVS + P40.3[7]Haut OgooueCHIKVp45.2–62.3[99]DENVP12.2[99]Ogooue LoloCHIKVP28.7[99]Woleu NtemCHIKVP0.5[18]nationwideCHIKVP, P35.6–86[96,127]DENVP, P0.2–94.8[96,105]212–220 villagesDENVS0.5[128]RVFVS3.3[112]CameroonEastZIKVS7.6[56]LittoralCHIKVS12.6–59.4[36,37]DENVS + P + R, S + P + R, P, P, S + R, R + S, R + P3.9–68.3[37,41,42,43,44,45,47]ZIKVS + P + R, S10–26.2[37,56]SouthDENVP, S + P0.5–14.28[18,129]NorthYFVS25.5[54]Far NorthDENVS + P + R, S + P + R6.7–14.36[44,45]ZIKVS2–4.8[56]AdamawaDENVS + R, R4.7–6.89[47,48]ZIKVS2[56]WestCHIKVS + P15.7[46]DENVS + R, S +R, S + P, P + R6.14–14.36[44,45,46,47]CenterCHIKVS, S + P + R, S + P3–59.4[36,37,39]DENVS + P + R, S + R, S + R, P + R, S + P3–45.45[37,39,44,45,47]ZIKVS3.3[56]North WestCHIKVS51.4[9]South WestCHIKVS, S4–63[6,38]ZIKVS11.4[6]DENVS, S2.5–74[6,38]YFVS4–72[38]WNVS3–82[38]Democratic Republic of CongoMatadiCHIKVS + P + R83.2[78]KisanganiWNVS66[81]CHIKVS34[81]DENVS3[81]RVFVS4[45]kinshasaDENVS, R, P, S + P + R, S + P0.4–8.1[73,80,85,86,88]YFVS, S + P6.0–73[73,77]CHIKVS, R, S + P, S, P, P0.1–49.7[73,80,82,88,130,131]Sud-UbangiZIKVS3.5%[73]Republic of the CongoBrazzavilleCHIKVP, S + P11.7–71[115,132]Pointe-NoireP, S + P-[15,114]-ZIKVS1.8[117]-DENVS + P-[24]AngolaLuandaDENVP, P + R, S + P11.1–94.4[8,31,133]CHIKVP, P7[8,134]13 provincesYFVS70[135]Equatorial GuineaBataCHIKVP1.1–33.3[14]ChadN’DjamenaYFVS + P, S + R0.28[24,70]RVFVP4[72]DENVS + P-[24]Central African RepublicBanguiYFVS + P6.5[136]RVFVP1.9–16.7[65]CHIKVP-[60]Chikungunya virus (CHIKV); dengue Virus (DENV); Zika Virus (ZIKV); yellow fever Virus (YFV); Rift Valley fever virus (RVFV); West Nile virus (WNV). (-) not available; RNA detection (P); rapid diagnostic test (R); serology (S); serological and RNA detection in the same paper (S + P); serology, RNA detection, and rapid diagnostic test (S + P + R). The order of tests separated by commas in the diagnosis column corresponds to the references in the square brackets of the references column.

### 3.6. The Mosquito Fauna of the Central African Region from 1993 to 2023

After a thorough online search, we documented >248 species of mosquitoes, regrouped under 15 genera in decreasing order of magnitude as follows: *Anopheles* (*n* = 100), *Culex* (*n* = 56), Aedes (*n* = 52), *Uranotaenia* (*n* = 12), *Coquillettidia* (*n* = 10), *Eretmapodites* (*n* = 5), *Ficalbia* (*n* = 4), *Mansonia* (*n* = 2), *Mimomyia* (*n* = 2), Ochlerotatus (*n* = 2), *Lutzia* (*n* = 1), Orthopodomyia (*n* = 1), and Verrallina (*n* = 1), but *Finlayas* and *Toxorhinchites* species were unidentified (Figure 4, Table 2).

From the current study, it is clear that the genus *Culex* constituted the most diverse group with the highest species frequency, and this could be accounted for by their high adaptability in different agroecological settings and their ability to colonise diverse microhabitat-types for breeding, survival, and proliferation. The genus *Aedes* was the second-most-frequent group, with invasive species such as *Ae. aegypti* and *Ae. albopictus* that codominated rural and urban spaces [111]. These two species were frequently identified in all countries presenting data on potential mosquito fauna of arboviruses. Although *Ae. aegypti* has been frequently reported [50,137], *Ae. albopictus* usually occurs in higher proportions in the Central African Region [111,131] (Table 3). Despite the fact that studies reporting circulating arboviruses in mosquitoes in the Central African region are scant, papers published elsewhere in West Africa, particularly in the Ivory Coast, reported very low infection rates in local *Ae. aegypti* [138]. The occurrence of different species in countries of this region could be an indication of an eventual spillover of arboviruses [139].
microorganisms-12-00004-t002_Table 2Table 2Mosquito fauna reported in some Central African countries from January 1993 to June 2023.SpeciesCountriesReferencesCMRRoCGabCARDRCEGAngCha*Aedes aegypti*++++++++[10,20,29,34,40,46,50,51,52,53,57,58,68,69,78,95,96,99,102,103,104,108,109,111,114,115,118,131,137,139,140,141,142,143,144,145,146,147,148,149,150,151,152,153,154,155,156,157,158,159,160,161,162,163,164,165,166,167,168]*Aedes abnormalis*


+



[67]*Aedes africanus*+

+
+

[9,10,40,67,68,95,99,139,141,148,150,168,169,170,171,172]*Aedes albopictus*++++++++[9,10,15,20,29,52,53,57,58,69,78,83,95,96,102,111,112,113,114,115,116,117,118,119,120,121,122,123,124,125,126,127,128,129,130,131,132,133,134,135,136,137,139,141,142,143,144,145,147,148,149,152,161,162,163,166,167,170,172,173,174,175]*Aedes alternans*






+[69]*Aedes argenteopunctatus*+

+



[10,40,139,141,171]*Aedes australis*






+[69]*Aedes caspiua*






+[69]*Aedes centropunctatus*


+



[141]*Aedes cinerus*






+[69]*Aedes circum*+

+



[10,40,58,139]*Aedes circumluteocus*+
++



[40,139,171,172,176]*Aedes contigus*+






[53,170]*Aedes cumminsii*+

+



[10,141]*Aedes dalzieli*+
++



[58,141,172]*Aedes dendrophilus*







[10]*Aedes domesticus*+






[10,139]*Aedes dufouri*






+[69]*Aedes fraseri*+






[40,139,171]*Aedes flavifrons*






+[122]*Aedes fowleri*+

+


+[58,67,69,141]*Aedes furcifer*

+




[172]*Aedes gibinsis*+






[139,171]*Aedes haworthi*


+



[10]*Aedes ingrani*+






[10]*Aedes irritans*

+




[159]*Aedes longipalpis*

++



[10,172]*Aedes luteocephalus*


+



[10,68,141]*Aedes mcintoshi*+

+



[58,141]*Aedes metallicus*+






[40,139]*Aedes minutus*

+




[172]*Aedes mixtus*


+



[141]*Aedes mucidus*+






[58]*Aedes multiplex*






+[69]*Aedes nigricephalus*

+




[159]*Aedes notoscriptus*






+[69]*Aedes ochraceus*+






[58]*Aedes opok*


+



[10,67]*Aedes palpalis*+






[10,53,67]*Aedes polynesiensis*






+[69]*Aedes procax*






+[69]*Aedes simpsoni*+
++



[10,20,40,53,57,68,96,103,118,139,141,145,170]*Aedes simulans*

+




[177]*Aedes soleatus*+






[40,139]*Aedes stockesi*


+



[10]*Aedes subargenteopunctatus*


+



[10]*Aedes tarsalis*+






[10,40,67,139,154,171]*Aedes vexans*+
+



+[69,149,157,172]*Aedes vigilax*

+



+[69,172]*Aedes vittatus*++++



[10,58,118,137,141,145,154,172]*Aedes vittiger*






+[69]*Aedes wellmani*+






[40,139]*Anopheles annuliotus*


+



[10,171]*Anopheles annulipes*






+[69]*Anopheles arabiensis*+


+
++[69,178,179,180]*Anopheles ardensis*
+

+
+
[178]*Anopheles argenteolobatus*



+
+
[178]*Anopheles austenii*



+
+
[178]*Anopheles azevedoi*





+
[178]*Anopheles bambusae*


+



[10]*Anopheles barberellus*
+

+
+
[178]*Anopheles berghei*



+


[178]*Anopheles bervoetsi*+


+

+[178]*Anopheles brohieri*++
+



[178]*Anopheles brunnipe*++

+++
[178,181]*Anopheles buxtoni*+






[178]*Anopheles caliginosus*



+
+
[178]*Anopheles carnevalei*+
+

+

[178,182,183]*Anopheles caroni*
++




[178]*Anopheles christyi*+


+

+[178]*Anopheles cinctus*+++++++
[141,178]*Anopheles cinerus*
+




+[178]*Anopheles claviger*






+[69]*Anopheles coluzzii*++++++++[69,149,157,178,180]*Anopheles confusus*



+


[178]*Anopheles cconcolor*+


+
+
[178,180]*Anopheles coustani*+++++
++[10,69,141,178,179,180]*Anopheles cydippis*++

+
++[178]*Anopheles dureni*
+

+
+
[178]*Anopheles deemingi*+






[178]*Anopheles demeilloni*++

+
+
[178]*Anopheles distinctus*



+
+
[178]*Anopheles dthali*






+[178]*Anopheles domicolus*+

++


[178]*Anopheles dualaensis*+






[178]*Anopheles eouzani*+
+




[178]*Anopheles faini*

+
+


[177,178]*Anopheles flavicosta*+

+

+
[141,178]*Anopheles freetownensis*++
+



[178]*Anopheles fontenillei*

+




[178]*Anopheles fuscivenosus*





+
[178]*Anopheles funestus*++++++++[10,33,67,69,141,146,154,177,178,179,180,181,182]*Anopheles gambiae*++++++++[10,20,32,33,67,69,96,102,109,111,118,137,139,141,142,143,144,145,146,147,150,154,157,159,164,170,178,179,180,181,182,184]*Anopheles garnhami*



+


[178]*Anopheles gabonensis*

+




[178]*Anopheles gibbinsi*
+
++


[178]*Anopheles hamoni*
+





[178]*Anopheles hancocki*+++
+


[178,181]*Anopheles hargreavesi*+++++


[178]*Anopheles charperi*





+
[178]*Anopheles hyrcanus*






+[69]*Anopheles implexus*+++++
+
[10,178,183]*Anopheles jebudensis*+++
+
+
[177,178]*Anopheles keniensis*



+


[178]*Anopheles kingi*



+


[178]*Anopheles leesoni*++
++++
[178]*Anopheles listeri*





+
[178,180]*Anopheles lloreti*




+

[178]*Anopheles longipalpis*++

+
+
[178]*Anopheles maculipalpis*+
+++
+
[178]*Anopheles maculipennis*



+

+[69,171]*Anopheles marshallii*+++++
+
[176,177,178,183]*Anopheles melas*+++++++
[32,49,167,178,180,182]*Anopheles meraukensis*

+




[172]*Anopheles millecampsi*



+


[178]*Anopheles mortiauxi*



+


[178]*Anopheles moucheti*++++++

[147,149,170,178,181,182,183]*Anopheles mousinhoi*+


+


[178]*Anopheles multicinctus*+

++


[178]*Anopheles namibiensis*+






[178]*Anopheles natalensis*+++++
+
[10,177,178]*Anopheles nili*++++++++[10,141,170,177,178,179,180,181,182]*Anopheles njombiensis*



+
+
[178]*Anopheles obscurus*+++++++
[176,178]*Anopheles okuensis*+






[178]*Anopheles ovengensis*+



+

[178,182]*Anopheles paludis*+++++
+
[10,170,172,176,178,181]*Anopheles pharaoensis*+
+++
++[69,109,141,178,179,180]*Anopheles pretoriensis*+++++
+
[178]*Anopheles rageaui*++

+


[178]*Anopheles rhodesiensis*+++++
+
[178]*Anopheles rodhaini*



+


[178]*Anopheles rivulorum*++

+
+
[178]*Anopheles rivulorum-like*





+
[178]*Anopheles ruarinus*





+
[178,180]*Anopheles rufipes*+++++
++[67,141,178,179,180]*Anopheles schwetzi*

+
+
+
[177,178]*Anopheles seydeli*



+


[178]*Anopheles smithii*+
++
+

[177,178]*Anopheles somalicus*+






[178]*Anopheles squamosus*+++++
++[58,178,179,180]*Anopheles symesi*



+


[178]*Anopheles tchekedii*





+
[178]*Anopheles tenebrosus*+
+++
++[147,149,157,176,178]*Anopheles theileri*

+
+
+
[178]*Anopheles vanhoofi*
+

+


[178]*Anopheles versus*



+


[177]*Anopheles vinckei*

+




[33,178]*Anopheles walravensi*



+
+
[178]*Anopheles wellecomei*+
+++
++[69,178,179]*Anopheles ziemmani*+++++
++[67,141,147,178,179,180]*Anopheles zombaensi*

+




[177]*Coquelettidia* spp.+

+



[10,141,170]*Coquelettidia annettii*+






[139]*Coquelettidia aurites*

+




[176,177]*Coquelettidia cristata*


+



[10]*Coquelettidia fraseri*


+



[10]*Coquelettidia maculipennis*+






[40,139]*Coquelettidia microannulata*

+




[177]*Coquelettidia pseudoconopas*

++



[10,177]*Coquelettidia richiardii*






+[69]*Coquelettidia versicor*

+




[177]*Coquelettidia xanthogaster*






+[69]*Culex albiventis*+
+




[40,185]*Culex annulioris*+
++



[10,40,141,172,177,178]*Culex annulirostries*






+[69]*Culex antenatus*+
+




[40,58,145,150,159,185]*Culex andersoni*

+




[177]*Culex argenteopunctatus*+






[185]*Culex australicus*






+[69]*Culex cinerus*+
++



[141,142,159,176,177,185]*Culex cinerellus*+
++



[111,159,177,185]*Culex duttoni*+
+

+

[40,53,67,95,111,139,145,147,149,150,154,157,170,171,172,185,186,187]*Culex decens*+
+




[67,95,109,111,142,145,146,150,157,159,176]*Culex eouzani*+






[185]*Culex fatigans*+






[188]*Culex guiarti*+






[185]*Culex horridus*+






[185]*Culex individiosus*+






[139]*Culex insignis*+





+[69,185]*Culex macfie*+






[185]*Culex muspratti*+






[185]*Culex musarum*+






[185]*Culex simpliciforceps*+






[185]*Culex moucheti*+






[40,53,139,170,171,185]*Culex modestus*






+[69]*Culex molestus*






+[69]*Culex neavei*+
+



+[67,172,185]*Culex nebulosus*+
++



[141,159,177,185]*Culex orbostiensis*






+[69]*Culex ornothoracic*+






[139,171,185]*Culex perexiguus*






+[69]*Culex perfuscus*+






[139,145,154,170,185]*Culex perfidiosus*+

+



[10,67,141,142,154,170,185]*Culex pipiens*+





+[69,139,150,186]*Culex phillipi*+






[40,185]*Culex poicilipes*+

+



[10,69,147,150,157]*Culex poecilipes*+
+




[149]*Culex pruina*+






[40,170,185]*Culex quasiguiarti*

+




[177]*Culex quinquefasciatus*++++
+
+[10,20,40,53,58,69,95,96,109,111,118,141,142,143,145,146,147,149,157,159,164,167,170,172]*Culex rubinotus*+
+




[176,177]*Culex rima*+
+



+[69,159,177,185]*Culex sitiens*






+[69]*Culex schwetzi*+






[185]*Culex semibrunneus*+






[177,185]*Culex simpsoni*+
+




[149,150,177]*Culex sunyaniensis*+






[185]*Culex subaequali*+






[185]*Culex tigripes*+++
+


[10,40,53,95,109,137,141,142,145,146,150,154,159,171]*Culex trifilatus*+
+




[139,177]*Culex univittatus*+
+




[40,139,150,170,171,177]*Culex watti*

+



+[69,177]*Culex wiggleworthi*+






[40,139,170,171,185]*Culex taufliebi*

+




[111]*Culex thalassius*+






[185]*Culex theileri*

+




[177]*Culex trifoliatus*+
+




[111,139,185]*Culex umbripes*

+




[111,185]*Eretmapodites* spp.+
++
+

[40,53,139]*Eretmapodites quinquevittatus*+
+

+

[20,95,145]*Eretmapodites chrysogaster*+
+




[10,40,139,171,177]*Eretmapodites grahami*

+




[177]*Eretmapodites inornatus*

++



[10,111,172]*Eretmapodites plioleucus*+






[139]*Ficolbia* spp.


+



[40]*Ficalbia Flavopicta*+






[139]*Ficalbia malfeyi*

+




[176]*Ficalbia mediolineata*

+




[159]*Ficalbia uniformis*

+




[177]*Finlayas* spp.

+




[177]*Lutzia tigripes*+
++


+[40,67,69,111,139,142,170,177,186]*Mansona africana*+
++
+

[10,20,58,95,96,141,143,159,170,172,176]*Mansona uniformis*+
++


+[10,20,58,69,96,102,109,141,143,159,170]*Mimmonyia* spp.+






[40]*Mimmonyia flavopicta*+






[139]*Mimmonyia plumosa*

+




[177]*Ochlerothatus rusticus*






+[69]*Ochlerothatus excrucians*






+[69]*Orthopodomyia reunionensis*






+[69]*Uranotaenia* spp.+
++



[40]*Uranotaenia bilineata*+
+




[139,171,176,177]*Uranotaenia cavernicola*

+




[176,177]*Uranotaenia nigromaculata*

+




[176,177]*Uranotaenia nigripes*

+




[177]*Uranotaenia caliginosa*

+




[176]*Uranotaenia caeruleocephala*

+




[176]*Uranotaenia machadoi*

+




[176]*Uranotaenia pallidocephala*

+




[176]*Uranotaenia balfoui*

+




[176]*Uranotaenia chorleyi*

+




[176]*Uranotaenia alboabdominalis*

+




[176]*Uranotaenia mashonaensis*+
+




[139,159,176,177]*Verralina funerea*






+[69]*Toxorhinchites* spp.+






[40,53]+: presence of species. Ang: Angola; CMR: Cameroon; Cha: Chad; Ga: Gabon; RoC: Republic of the Congo; CAR: Central African Republic; DRC: Democratic Republic of Congo; EG: Equatorial Guinea.


### 3.7. Arboviruses and Associated Mosquito Vectors in the Central African Region from 1993 to 2023

The spillovers of arboviruses of public health importance have reportedly been associated with mosquito genera of medical and zoonotic importance such as *Aedes*, *Culex, Anopheles*, etc. The vector competence of *Ae. aegypti* [139,189,190,191] and *Ae. albopictus* [190,192] in the transmission of major arboviruses has been well-documented. The vector competence and the association of some *Aedes* spp. as potential vectors of important viruses is still unknown. For the *Culex* mosquitoes already identified in studies of the Central African region, they have been frequently reported to be associated with zoonotic arboviruses such as RVFV and WNV [19,188,191,193]. Of the *Culex* spp. involved in the spread of zoonotic arboviruses, *Culex pipiens* was reported to be associated with CHIKV [19,193,194,195]. The genus *Anopheles* has also been reported to be associated with some medically important arboviruses, such as YFV (*An. gambiae* s.l.) [196], ZIKV (*An. moucheti*) [194], and CHIKV (*An. funestus*) [191]. Moreover, other mosquitoes that are not well-known in the region are associated with arboviruses and include *Eretmapodites* spp. (ZIKV) [40,191], *Coquelettidia* spp. (YFV and RVFV) [189,191,193], and *Lutzia tigripes*, which was reported to be potential vector of YFV, DENV, and WNFV [139,176,191]. The occurrence of these competent and potential vectors in the different countries of this subregion of Africa could indicate the risk of spillover of the different viruses highlighted in Table 3. Therefore, vector competence studies and transmission studies, together with clinical diagnosis, are required in order to keep track of the patterns of occurrence and spread of arboviruses of medical and zoonotic importance in the Central African region.
microorganisms-12-00004-t003_Table 3Table 3Arboviruses and associated mosquito vectors in the Central African region from January 1993 to June 2023.SpeciesVirusDiagnosisReferences*Ae. aegypti*DENVV, V, V, P, V[190,191,197,198,199]CHIKV, V, V, P, V, V, P + V, V, V[190,191,197,198,199,200,201,202,203]ZIKVV, V, V, V, P, V[149,190,191,197,198,199]YFVV, V, V, P, V, V[190,191,197,198,199,204]RVFVV + P, V, V, V[59,191,193,199]WNVP[194]Ross RiverV, V, V, P[190,191,197,198]Murray ValléeV, V, V, P[190,191,197,198]WesselsbronV, V, V + P[191,200,201]BakankiV, P[191,205]O’nyong NyongV, V[191,202]*Ae. africanus*CHIKVV, V, V, V, V[67,191,204,206]WNVV, V, P, V[67,191,205,206]YFVV, V, V, V[67,191,204,206]ZIKVV, V, P, V, V, V[67,148,172,191,204,206]RVFVV, V, P[67,204,205]BozoV, P[67,205]BoubouiV, V, P, V[67,172,191,206]BabankiV, V, P + V[191,200,201]UgandaV, V, P + V, V[191,200,201,206]WesselsbronV, V, V, P + V[67,191,200,201]OrungoV, P[67,205]MiddelburgV, V, P + V[191,200,201]SaboyaP, P[172,205]SemlikiV, V, P + V, V[191,200,201,206]YaoundéV, V, P + V, P[191,200,201,205]*Ae. albopictus*DENVV, P, V, V, V, P[29,96,190,192,197,198]CHIKVP, V, V, V, P, V + P[96,190,191,197,198,201]ZIKVV, V, V, V, P[148,190,191,197,198]YFVV, V, V, P[190,191,197,198]RVFVV + P[58]WNVV, V, P, V[190,191,197,198]UsutuV, V, P + V[191,200,201]Ross RiverV, V, P[190,198,198]Murray ValléeV, V, P[190,197,198]*Ae. argenteopunctatus*CHIKVV, V, P[191,200,201]YFVV, V, P[190,197,198]ZIKVV, V, P[190,197,198]SemlikiV, V, V, P[190,194,197,198]KedougouV[194]SimbuV[194]*Ae. bromelia*YFVV[204]*Ae. caballus*RVFVV, V, P, V[190,197,198,207]MidelburgV, V, P + V[191,200,201]WesselsbronV, V, V + P, V[191,200,201,204]*Ae. cordeleri*CHIKVV, V, V + P[191,200,201]*Ae. caspiua*UsutuV, V, P + V[191,200,201]*Ae. circumluteocus*RVFVV, V, V + P, V, V[58,191,193,200,201]WesselsbronV, V, P + V[191,200,201]PongolaV, V, P + V[191,200,201]BunyamweraV, V, P + V[191,200,201]NdumuV, V, P + V[191,200,201]SpondweniP, V, V, P + V[172,198,199,200]*Ae. cumminsii*RVFVV, V, P + V, V[189,200,201,208]SpondweniV, V, P + V[191,200,201]*Ae. dalzieli*CHIKVV, V[194,209]ZIKVP, V, V, V, P + V[172,191,194,200,201]RVFVV + P, P[58,181,210]MiddelburgV[194]NdumuV[194]KedougouV, V, P, P, V[172,191,194,200,201]WesselsbronP, V[172,194]BunyaweraV[194]ShokweV[194]SimbuV[194]PongolaV[194]ZingaV[194]*Ae. dentatus*YFVV[194]RVFVV, V, P + V[191,200,201]*Ae. domesticus*BunyamweraV, V, V[139,199,203]*Ae. furcifer/taylori*CHIKVV, V, V, P + V, V, V[189,191,200,201,209,211]ZIKVP, V, V, V, V + P[172,191,194,200,201]YFVV, V, V, V + P[191,194,200,201]BunyamweraV[194]BoubouiV[194]BwambaV, V, P + V[191,200,201]*Ae. longipalpis*UgandaP, V, V, P + V[172,191,200,201]*Ae. luteocephalus*CHIKVV, V, V, V, V + P, V[189,191,194,200,201,204]YFVV, V, V, V + P, V, V[191,194,200,201,204,209]ZIKVV, V, V + P, V[191,194,200,201]DENVV, V[194,204]BunnyamweraV, V, P + V[191,200,201]*Ae. mcintoshi*CHIKVV, V, P + V[191,200,201]RVFVV + P, V[58,193]NdumuV, V, V + P, V[191,200,201]PongolaV, V, V + P[191,200,201]WesselsbronV, V, V + P[191,200,201]BabankiV, V, V, V + P[191,192,200,201]NgariV, V, P + V[191,200,201]BunyamweraV, V, P + V[191,200,201]*Ae. metallicus*YFVV, V, V, V + P, V, P[189,191,200,201,204,205]ZIKVV, V, V + P, P[191,200,201,205]*Ae. minutus*ZIKVV, V, P + V[191,200,201]NdumuV[194]KedougouP, V, V, V, V + P[172,191,194,200,201]WesselsbronV[194]*Ae. neoafricanus*CHIKVV, V, P + V[191,200,201]YFVV, V[194,204]*Ae. ochraceus*RVFVV + P[58]NdumuV, V, P + V[191,200,201]BabankiV, V, P + V[191,200,201]*Ae. opok*CHIKVV, V, V[67,189,206]YFVV, V, V, V + P, V, V[67,191,200,201,204,206]WNVP[205]ZIKVV, V, V, V + P, V, V[67,191,200,201,204,206]BoubouiV, V[67,204]OrungoV[67]WesselbronV[67]BozoV[67]MiddelburgP[205]SaboyaP[205]SemenikiV[206]YaoundéP[205]*Ae. palpalis*RVFVV, V[67,193]MiddelburgV, V[199,212]SimbuV[67]*Ae. simpsoni*CHIKVV, V, P[139,189,205]YFVV, V, V + P, V[191,200,201,204]BabankiV, V, V + P[191,200,201]NgariV, V, V + P[191,200,201]*Ae. tarsalis*ZIKVV, V, V + P[191,200,201]PataV[67]BunyamweraV[212]MiddelburgV[212]PangolaV[67]KedougouV, V, V, V + P[67,191,200,201]WesselbronV[67]*Ae. tricholabic*NdumuV, V, V + P[191,200,201]PongolaP[192]BunyamweraP[212]NgariP[212]*Ae. vexans*wesselsbronP[172]*Ae. vigilax*EdgeP[172]*Ae. vittatus*CHIKVV, V, V, V + P[189,191,200,201]RVFVV + P[58]YFVV, V, V, V + P[191,194,200,201]ZIKVV, V, V, V + P[191,194,200,201]SindbisV[194]MiddelburgV[67]SemlikiV[194]WesselsbronV[67]BunyamueraV[194]SimbuV, V[67,194]PongolaV[194]SaboyaP[172]*Ae. abnormalis, Ae. alternans, Ae. australis, Ae. cinerus, Ae. centropunctatus, Ae. contigus, Ae. dendrophillus, Ae. dufouri, Ae. fraseri, Ae. flavifrons, Ae. fowleri, Ae. gibinsis, Ae. haworth, Ae. ingrani, Ae. irritans, Ae. mixtus, Ae. mucidus, Ae. multiplex, Ae. simulans, Ae. soleatus, Ae. stockesi Ae. polynesiensis, Ae. procax Ae. nigricephalus, Ae. notoscriptus, Ae. vittiger, Ae. subargenteopunctatus, Ae. wellmani*n.a
/*An. brohieri*SindbisV[194]*An. coustani*CHIKVV[194]BwanbaV, V, V + P[191,200,201]*An. funestus*CHIKVV, V, V, V + P[191,194,200,201]WNVV[67]NyandoV, V, V + P, V[67,191,200,201]NgariV, V, V + P[191,200,201]BwambaV, V, V + P, V[67,191,200,201]BunyamweraV, V, V + P[191,200,201]O’nyong NyongV, V, V + P[191,200,201]PongolaV[194]TataguineV[67]*An. gambiae*YFVV, V, V, V, V + P, V[139,189,191,200,201,204,212]ZIKVV, V, V + P[191,200,201]IleshaV, V, V, V + P[67,191,200,201]BwambaV, V, V, V + P[191,194,200,201]O’nyong NyongV, V, V + P[191,200,201]MiddelburgV, V[67,213]TataguineV[194]OrungoV[67]*Ae. mercaukensis*EdgeP[172]*An. moucheti*ZIKVV[194]*An. nili*TataguineV[194]*An. paludis*BoubouiP, V, V, V + P[172,191,200,201]*An. maculipennis*UsutuV, V, V + P[191,200,201]*An. annuliotus, An. annulipes, An. arabiensis, An. ardensis, An. argenteolobatus, An. austenii, An. azevedoi, An. bambusae, An. barberellus, An. berghei, An. bervoetsi, An. brunnipe, An. buxtoni, An. caliginosus, An. carnevalei, An. caroni, An. christyi, An. cinctus, An. cinerus, An. claviger, An. coluzzii, An. confusus, An. concolor, An. cydippis, An. dureni, An. deemingi, An. demeilloni, An. distinctus, An. dthali, An. domicolus, An. dualaensis, An. eouzani, An. faini, An. flavicosta, An. freetownensis, An. fontenillei, An. fuscivenosus, An. garnhami, An. gabonensis, An. gibbinsi, An. hamoni, An. hancocki, An. hargreavesi, An. harperi;An. hyrcanus, An. jebudensis, An. keniensis, An. kingi, An. leesoni, An. listeri, An. lloreti, An. longipalpis, An. maculipalpis, An. marshallii, An. melas, An. millecampsi, An. mortiauxi, An. mousinhoi, An. multicinctus, An. namibiensis, An. natalensis, An. njombiensis, An. obscurus, An. okuensis, An. ovengensis, An. pharaoensis, An. pretoriensis, An. rageaui, An. rhodesiensis, An. rodhaini, An. rivulorum, An. rivulorum-like, An. ruarinus, An. rufipes, An. schwetzi, An. seydeli, An. smithii, An. somalicus, An. squamosus, An. symesi, An. tchekedii, An. tenebrosus, An. theileri, An. vanhoofi, An. versus, An. vinckei, An. walravensi, An. wellecomei, An. ziemmani, An. zombaensis*n.a//*Coquelettidia.* spp.YFVV[189]*Co. fuscopennata*SindbisV, V, V, V + P[191,194,200,201]*Co. aurites*UsutuV, V, V + P[191,200,201]*Co. annettii, Co. cristata, Co. fraseri, Co. maculipennis, Co. microannulata, Co. pseudoconopas, Co. richiardii, Co. versicor, Co. xanthogaster*n.a
/*Cx. albiventis*ArumowotV, V, V + P[191,200,201]NtayaV[212]*Cx. annulirostris*KameseP, V, V, V + P[172,191,200,201]EdgeP[172]*Cx. antenatus*RVFVV + P, V[58,193]ArumowatV[191,200,201]*Cx. cinerus*M’PokoV[194,212]NtayaV[212]*Cx. duttoni*UgandaP[172]wesselsbronP[172]*Cx. decens*SindbisV[194]UsutuV[194]KameseV[67]NyandoV[67]*Cx. individiosus*SindbisV[194]*Cx. ingrani*BagazaV, V, V + P[191,200,201]*Cx. moucheti*NtayaV, V[139,212]*Cx. modestus*WNVV, V, V + P[191,200,201]*Cx. neavei*WNVV, V, V + P[191,200,201,209]UsutuV, V, V + P[191,200,201]SindbisV, V, V + P[191,200,201]SpondweniP, V, V, V + P[172,191,200,201]*Cx. nebulosus*YaoundéV, V, V + P[191,200,201]NtayaV[212]*Cx. perfuscus*WNVV, V[67,194]SindbisV[194]BagazaV, V, V, V + P[67,191,200,201]UsutuV, V, V, V + P[191,194,200,201]M’PokoV[67]WesselsbronV[67]*Cx. pipiens*CHIKVV[194]RVFVV[193]WNVV, P, V, V + P[191,195,200,201]SindbisP, V, V, V + P[191,194,200,201]BabankiP[194]UsutuV, V, V + P[191,200,201]*Cx. poicilipes*WNVV[209]*Cx. pruina*KameseV, V, V, V + P[67,191,200,201]BozoV[67]*Cx. quinquefasciatus*WNVV, V, V + P[191,200,201]RVFVV + P, V[58,193]WesselsbronV, V, V + P[191,200,201]*Cx. rubinotus*BanziP, V, V, V + P[172,191,200,201]arumowatV, V, V + P[191,200,201]WNVP[192]YaoundéP[192]NdumuP[192]*Cx. univittatus*WNVV, V, V + P[139,192,212]SindbisV, V, V + P[191,200,201]WesselsbronV, V, V + P[191,200,201]UsutuV, V, V + P[191,200,201]SpondweniV, V, V + P[191,200,201]NdumuV, V, V + P[191,200,201]BagazaV, V, V + P[191,200,201]*Cx. tarsalis*WNVP[195]*Cx. telesilila*SindbisV[194]NtayaV[212]*Cx. theileri*WNVV, V, V + P[191,200,201]*Cx. tigripes*NtayaV[212]MossurilV[67]*Cx. annulirostries, Cx. andersoni, Cx. argenteopunctatus, Cx. australicus, Cx. cinerellus, Cx. eouzani, Cx. fatigans, Cx. guiarti, Cx. horridus, Cx. insignis, Cx. macfiei, Cx. muspratti, Cx. musarum, Cx. simpliciforceps, Cx. molestus, Cx. nebulosus, Cx. orbostiensis, Cx. ornothoracic, Cx. perexiguus, Cx. perfidiosus, Cx. phillipi, Cx. poecilipes, Cx. quasiguiarti, Cx. rima, Cx. sitiens, Cx. schwetzi, Cx. semibrunerus, Cx. simpsoni Cx. sunyaniensis, Cx. subaequalis, Cx. watti, Cx. wiggleworthi, Cx. taufliebi, Cx. thalassius, Cx. trifoliatus, Cx. umbripes*n.a//*Er. quinquevittatus*ZIKVV, V, V + P[191,200,201]*Er. chrysogaster*SpondweniV[212]MiddelburgV[212]*Er. inornatus*ZIKVV, V, V, V + P[172,191,200,201]MiddellburgV[212]BoubouiP[172]*Er. grahami, Er. pliol*n.a
/*Ficolbia.* spp., *Fi. Flavopicta, Fi. malfeyi, Fi. mediolineata, Fi. uniformis*n.a
/*Finlayas.* spp.n.a
/*Lu. tiggipes*n.a
/*Ma. africana*RVFVV + P[58]MiddelburgV, V, V, V, V + P[67,191,200,201,212]WesselsbronV, V, V + P[191,200,201]SpondweniV, V, V + P[191,200,201]BanziP[172]*Ma. uniformis*ZIKVV, V, V, V + P[191,194,200,201]SpondweniV, V, V + P[191,200,201]BwambaV, V, V + P[191,200,201]O’nyong NyongV, V, V + P[191,200,201]NdumuV, V, V + P[191,200,201]*Mimmonyia.* spp., *Mi. flavopicta, Mi. plumosa*n.a//*Oc. rusticus, Oc. excrucians*n.a//*Or. reunionensis*n.a//*Ur. machadoi*n.a//*Uranotaenia.* spp., *Ur. bilineata, Ur. cavernicola, Ur. cavernicola, Ur. nigripes, Ur. caliginosa, Ur. caeruleocephala, Ur. pallidocephala, Ur. balfoui, Ur. chorleyi, Ur. alboabdominali, Ur. mashonaensis*n.a//*Ve. funerea*n.a//*Toxorhinchite.* spp.MosurilV, V[67,213]KameseV, V[67,213]Not available (n.a); virus isolation (v); P: RNA detection; virus isolation and RNA detection in the same study (V + P). The order of techniques used to show the vector transmission potential of the different mosquito species separated by commas in the diagnosis column refers and/or corresponds to the references in the square brackets of the references column.


Of the 164 eligible papers on the topic used in this study, we found that most of them were from Cameroon, followed by Gabon, and the fewest were recorded in Equatorial Guinea. The most commonly reported arboviruses to cause epidemics were chikungunya and dengue. The entomological records showed >248 species of mosquitoes in different studies related to arboviruses and regrouped under 15 genera, with *Anopheles* (*n* = 100 species), *Culex* (*n* = 56 species), and *Aedes* (*n* = 52 species) having the highest species diversity and broadest distribution. Three genera were rarely represented (with only one species), and these included *Orthopodomyia*, *Lutzia*, and *Verrallina*, but individuals of the genera *Toxorhinchites* and *Finlayas* were not identified up to the species level. We found that these two *Aedes* species were involved in major epidemics of the six medically important arboviruses, and other rare mosquito genera consisted of competent species and were associated with outbreaks of zoonotic arboviruses. These findings revealed the existing gaps in the epidemiological and entomological data of the various countries of the central African region, and the need for regular surveillance at the country level. There is need to focus research on arbovirus ecology and to establish the vector competence of other frequently identified mosquitoes in the region. Although few regional studies were documented, there is still a need to conduct a multicountry project on arboviruses and vectors in Central Africa in order to propose sustainable control measures.

## 4. Conclusions

The present study shows that the wide spread of competent mosquito vectors could lead to the spillover of medically important arboviruses in the region, presumably via the free movement of animals and people via porous borders. Although epidemiological studies were found, they were not regularly documented, and this also applies to vector competence and transmission studies. Future studies will consider raw data from technical, scientific, and administrative reports/archives (Ministry of Scientific Research and Ministry of Health) and unpublished information in dissertations (research institutions and Universities) that could enable the study to be more complete. A regional project, organised by the authors of this current work, entitled “Ecology of Arboviruses” (EcoVir) is underway in three countries (Gabon, Benin, and Cote d’Ivoire) to generate more comprehensive epidemiological and entomological field data on this topic. 

## Figures and Tables

**Figure 1 microorganisms-12-00004-f001:**
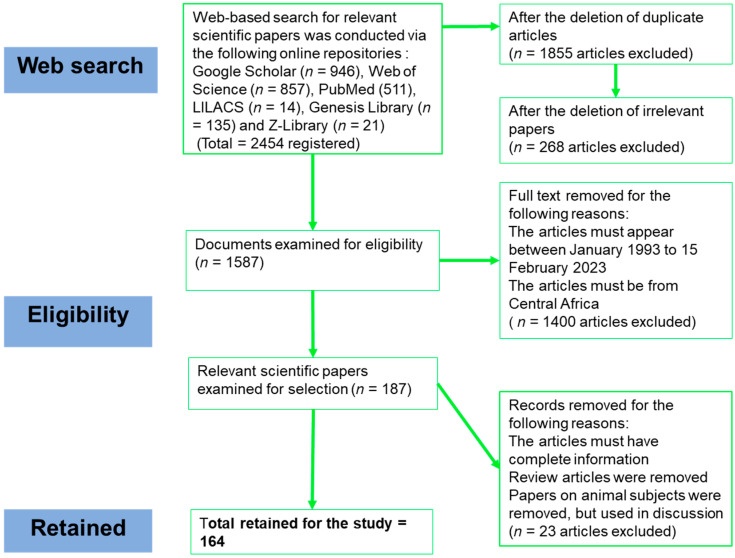
Flow chart of the steps in the reviewing process.

**Figure 2 microorganisms-12-00004-f002:**
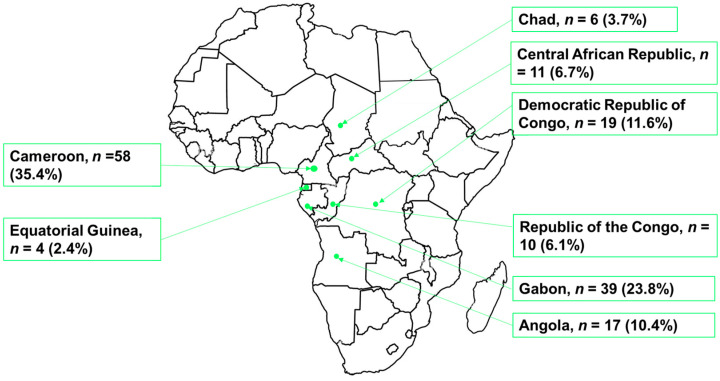
Number of articles retained by each country from 1993 to 2023.

**Figure 3 microorganisms-12-00004-f003:**
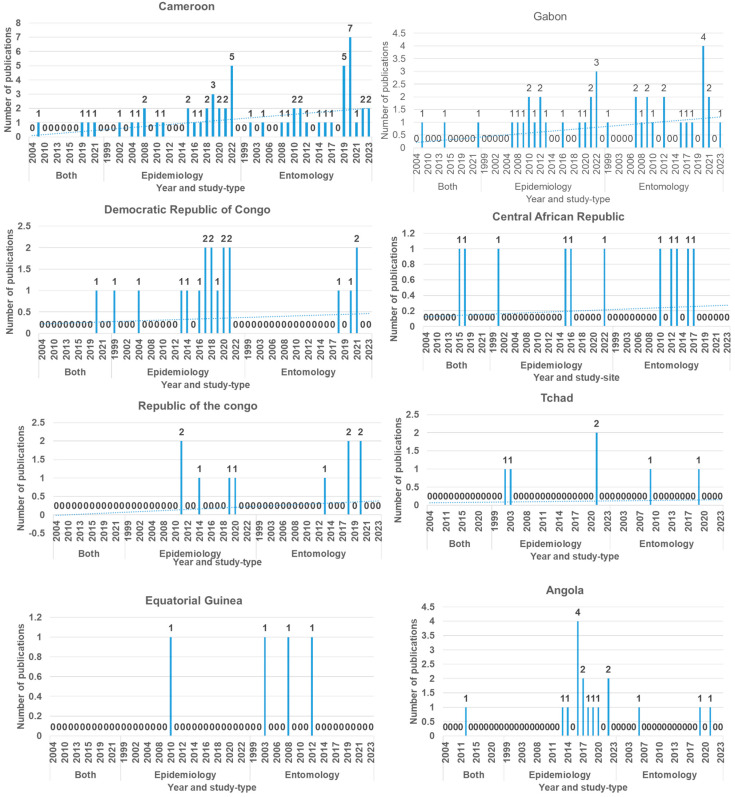
The periodic trend in the publication of entomological and epidemiological data by country.

**Figure 4 microorganisms-12-00004-f004:**
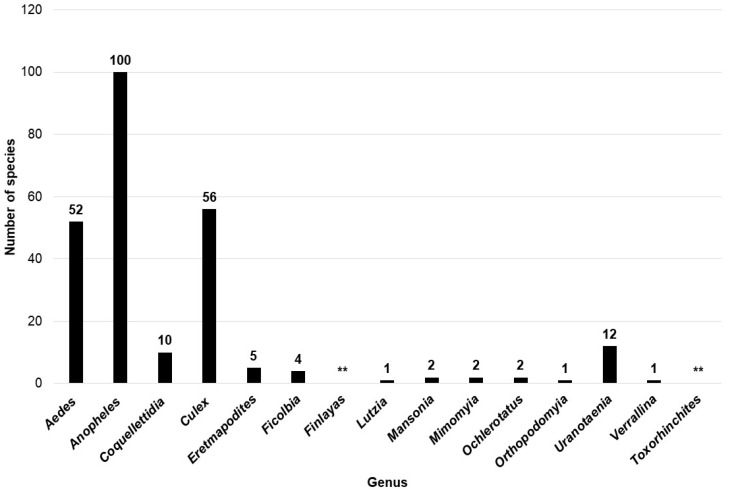
Genera of mosquitoes identified in the Central African subregion from January 1993 to June 2023; (**) unidentified species.

## Data Availability

Available upon request from the corresponding author.

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
