# Peer review of "Mosquito-Borne Arboviruses Occurrence and Distribution in the Last Three Decades in Central Africa: A Systematic Literature Review"

_microorganisms, 2023, doi:10.3390/microorganisms12010004_

Round 1
Reviewer 1 Report
see the file

need revision
Author Response
Reviewer 1
The willingness of author to collate arbovirus data collected in Central African Region for the last 30 years is a good idea and will be of great help for researchers. The decision to separate the 6 most studied viruses is also a good idea.
Thanks for the kind words
Major
M1. I suggest to identify the 164 studies that result from the reviewing process in the total list of references (n=191)
We have added a supplementary excel file with all the 164 articles.
M2. Table 3: the way the data are presented is quite useless; I recommend to
Replace lines dedicated to species for which no data is available by a sequence with comma
|
Ae dufouri |
n.a. |
|
|
Ae fraseri |
n.a. |
|
|
Ae flavifrons |
n.a. |
|
|
Ae fowleri |
n.a. |
|
By
|
Ae dufouri, Ae fraseri, Ae flavifrons, Ae fowleri |
n.a. |
|
This will help to save space since sometimes 20 lines are n.a.
Thanks for this orientation we have reorganized this table as indicated.
In the Virus column, list each viral species by line, the six viruses (CHIKV, DENV, ZIKV, WNV, YFV, RVFV can be acronymized to save space) such as indicated below
To make this table more informative, I believe that the type of evidence for the presence of the listed viruses should be described (V for virus isolation, R for RNA detection, S for serological evidence
|
Ae cumminsii |
|
Ref 175 |
Ref 188 |
|
|
Middleburg |
V |
|
|
|
Nkolbisson |
R |
|
|
|
RVFV |
|
S |
|
|
Shokwe |
|
R/S |
|
|
Sindbis |
V |
R |
|
|
Spondweni |
R/S |
V |
Revise the table to avoid redundancies: in the Ae africanus, Bouboui, Orungo and Bozo are listed twice: the same problem is found many times in the table: please correct
Thanks for the observation, the table has been cleaned to remove all double names as suggested.
Verify virus names: in the Ae aegypti, Encephalitis virus is listed: it does not exist, please correct
Thanks for the remark we have removed the name, it is true it does not exist.
The fact that Murray Valley virus is also listed should be checked or discussed since this is an Oceanian virus and his presence in Africa is awkward. Other viruses rarely described in Africa should also be included in the discussion.
The apparition of strange viruses in Africa in the past decades is not surprising due to increased mobility, urbanization and climate change that favors the expansion of mosquito biotopes and the spread of arboviruses.
Minor
Figure 2. for easier reading, I suggest to group the Country name with the number of studies such as 3. Chad, n=6 (3.7%) instead of having the country names in a separate rectangle.
Thanks for this important suggestion, we have corrected the figure.

Reviewer 2 Report
The review titled "Mosquito-borne arboviruses occurrence and distribution in the last three decades in central Africa: a systematic literature review" is an interesting work. This reviewer has following suggestions:
1. Introduction is too short and omitted crucial background information on arboviruses in Central African countries. Authors should include relevant backaground information highlighting at least some previous work on arboviruses in the Central African region.
2. L55: "Yellow Fever (YF) alone is of major public health concern". Is it worldwide?
3. L56: Spell out these acronyms at first use in the text.
4. L65-66: "and in some countries of this region, they are main drivers of arboviruses spillovers". Needs more specific information with citations.
5. L72: Why only for last 30 years? Any justification for that? Needs clarification on the chronology of the disease in the region.
6. Overall, Introduction needs more background information related to the region selected, with citations.
7. Table 2: The + symbol should be identical.
8. Before conclusions, there should be a summary paragraph in the Results and Discussion section providing a brief overview of the findings and their significance.
9. Most of the text written in Conclusions section can be taken to the Results and Discussion section. Conclusions should be revised with a clear and shortened message, including recommendations.
10. In Results and Discussion section, the authors focussed more on the availability of number of published papers from different Central African countries. In my opinion, authors should have focussed on the significance of these arboviruses in humans including the information on vectors. Mechanisms of dispersal for various arboviruses should be included, if possible, with graphics. Though tables provide significant information but they are not discussed properly in the text.
11. A section can be added highlighting on the detection techniques of arboviruses and the infrastructure available in these Central African countries for successful detection of arboviruses. Lack of papers on arboviruses from a given country does not rule out the possibility of the presence of arboviruses in that country. Rather it is the matter of importance given for research, availability of resources, and policies of that country. This is relevant for many Central and West African countries given the scarcity of resources. Authors should emphasize more on the techniques used for detecting arboviruses in these countries such as molecular detection versus serological? If PCR was used then which genes were targeted for various arbovirus detection? Please provide the information on primers and/or probes used for molecular detection.Genome sequences available for different arboviruses and their use in epidemiological understanding of their dispersal in the region would be useful. A phylogenetic tree for a given virus in the study region using previously reported sequences on the GenBank would be very useful in understanding their dispersal within the region. Information on vectors would be useful. Role of birds in West nile virus dispersal should also be considered in context of the migratory flyways that occur in Central Africa. Merely stating the published number of papers on various arboviruses from a given country may not provide useful information. For example, if Cameroon reported highest number of papers on arboviruses, is that because there is better surveillance for arboviruses in Cameroon and better infrastructure for research and arbovirus detection? Are there other factors that make Cameroon more prone for arbovirus infestation compared to other Central African countries? I am trying to say that authors need to be more critical in analyzing their work.
12. Overall, the topic chosen by authors is quite interesting but it needs a significant revision and additions to make it more useful for the readers.
English needs minor improvements at places.
Author Response
Reviewer 2
The review titled "Mosquito-borne arboviruses occurrence and distribution in the last three decades in central Africa: a systematic literature review" is an interesting work. This reviewer has following suggestions:
- Introduction is too short and omitted crucial background information on arboviruses in Central African countries. Authors should include relevant backaground information highlighting at least some previous work on arboviruses in the Central African region.
Thanks for this interesting comment. Relevant background information has been added.
- L55: "Yellow Fever (YF) alone is of major public health concern". Is it worldwide?
The Yellow Fever Virus is of major publich health concern in Africa and beyond.
- L56: Spell out these acronyms at first use in the text.
Thanks for the remark the names of the diseases have been written in full.
- L65-66: "and in some countries of this region, they are main drivers of arboviruses spillovers". Needs more specific information with citations.
Thanks for this observation we have added a reference from Cameroon.
- L72: Why only for last 30 years? Any justification for that? Needs clarification on the chronology of the disease in the region.
This has been justified. Again thanks for the remark, we started documenting information from 1993 because this period represented period of high epidemics all over Africa and to get more consistent information we decided to conduct a three decades review of literature on the topic.
- Overall, Introduction needs more background information related to the region selected, with citations.
We have added more literature to the introduction section as requested.
- Table 2: The + symbol should be identical.
Thanks for the remark it was a formatting error please accept our apologies. This has been corrected in the present version.
- Before conclusions, there should be a summary paragraph in the Results and Discussion section providing a brief overview of the findings and their significance.
Thanks for the remark, this section has been added.
- Most of the text written in Conclusions section can be taken to the Results and Discussion section. Conclusions should be revised with a clear and shortened message, including recommendations.
Thanks for the remark, this has been done
- In Results and Discussion section, the authors focussed more on the availability of number of published papers from different Central African countries. In my opinion, authors should have focussed on the significance of these arboviruses in humans including the information on vectors. Mechanisms of dispersal for various arboviruses should be included, if possible, with graphics. Though tables provide significant information but they are not discussed properly in the text.
We have added some intepretation
- A section can be added highlighting on the detection techniques of arboviruses and the infrastructure available in these Central African countries for successful detection of arboviruses. Lack of papers on arboviruses from a given country does not rule out the possibility of the presence of arboviruses in that country. Rather it is the matter of importance given for research, availability of resources, and policies of that country. This is relevant for many Central and West African countries given the scarcity of resources. Authors should emphasize more on the techniques used for detecting arboviruses in these countries such as molecular detection versus serological?
We created a column to indicate the diagnostic tests used and this was expanded in text.
If PCR was used then which genes were targeted for various arbovirus detection? Please provide the information on primers and/or probes used for molecular detection.Genome sequences available for different arboviruses and their use in epidemiological understanding of their dispersal in the region would be useful. A phylogenetic tree for a given virus in the study region using previously reported sequences on the GenBank would be very useful in understanding their dispersal within the region. Information on vectors would be useful. Role of birds in West nile virus dispersal should also be considered in context of the migratory flyways that occur in Central Africa. Merely stating the published number of papers on various arboviruses from a given country may not provide useful information. For example, if Cameroon reported highest number of papers on arboviruses, is that because there is better surveillance for arboviruses in Cameroon and better infrastructure for research and arbovirus detection?
I think no because there is still irregular surveillance and probably due to insufficient funds
Are there other factors that make Cameroon more prone for arbovirus infestation compared to other Central African countries? I am trying to say that authors need to be more critical in analyzing their work.
We have added more text about the techniques, presence of some arboviruses and vectors. Although most articles did not present detail information about the primer sequences and target genes, we managed to get such information from some papers.
- Overall, the topic chosen by authors is quite interesting but it needs a significant revision and additions to make it more useful for the readers.
Thanks for the kind words and the interesting remarks.

Round 2
Reviewer 2 Report
Thanks to authors for addressing all previous comments. No more technical comments from me. The English used at places can be significantly improved. I believe that the manuscript would immensely benefit from an extensive English editing.
I recommend authors perform extensive English editing which will improve the manuscript.